# Access to Care and Healthcare Quality Metrics for Patients with Advanced Genitourinary Cancers in Urban versus Rural Areas

**DOI:** 10.3390/cancers15215171

**Published:** 2023-10-27

**Authors:** Haoran Li, Kamal Kant Sahu, Shruti Adidam Kumar, Nishita Tripathi, Nicolas Sayegh, Blake Nordblad, Beverly Chigarira, Sumati Gupta, Benjamin L. Maughan, Neeraj Agarwal, Umang Swami

**Affiliations:** 1Division of Medical Oncology, University of Kansas Cancer Center, Westwood, KS 66205, USA; 2Division of Oncology, Internal Medicine, Huntsman Cancer Institute, University of Utah, Salt Lake City, UT 84112, USA; 3Department of Internal Medicine, University of Connecticut, Farmington, CT 06030, USA; 4Department of Internal Medicine, Wayne State University, Detroit, MI 48202, USA; 5Department of Internal Medicine, UT Southwestern Medical Center, Dallas, TX 75235, USA

**Keywords:** healthcare, disparity, cancer, genitourinary malignancy, rural, urban, population

## Abstract

**Simple Summary:**

This study aimed to compare the healthcare quality and results for patients with advanced genitourinary cancers from both rural and urban backgrounds, treated at Huntsman Cancer Institute in Utah. Even though urban residents had a median household income that was higher than rural patients and differences in insurance types, both groups had similar cancer characteristics when they began treatment. Importantly, the type of treatments received, including participation in clinical trials or specific cancer genetic tests, were the same for both groups. The survival outcomes for prostate, bladder, and kidney cancer were also similar for both rural and urban patients. This study’s findings suggest that when patients have access to specialized care, like that at a major cancer hospital, the differences in healthcare quality and outcomes between urban and rural patients can be reduced.

**Abstract:**

Compared to the urban population, patients in rural areas face healthcare disparities and experience inferior healthcare-related outcomes. To compare the healthcare quality metrics and outcomes between patients with advanced genitourinary cancers from rural versus urban areas treated at a tertiary cancer hospital, in this retrospective study, eligible patients with advanced genitourinary cancers were treated at Huntsman Cancer Institute, an NCI-Designated Comprehensive Cancer Center in Utah. Rural–urban commuting area codes were used to classify the patients’ residences as being in urban (1–3) or rural (4–10) areas. The straight line distances of the patients’ residences from the cancer center were also calculated and included in the analysis. The median household income data were obtained and calculated from “The Michigan Population Studies Center”, based on individual zip codes. In this study, 2312 patients were screened, and 1025 eligible patients were included for further analysis (metastatic prostate cancer (*n* = 679), metastatic bladder cancer (*n* = 184), and metastatic renal cell carcinoma (*n* = 162). Most patients (83.9%) came from urban areas, while the remainder were from rural areas. Both groups had comparable demographic profiles and tumor characteristics at baseline. The annual median household income of urban patients was $8604 higher than that of rural patients (*p* < 0.001). There were fewer urban patients with Medicare (44.9% vs. 50.9%) and more urban patients with private insurance (40.4% vs. 35.1%). There was no difference between the urban and rural patients regarding receiving systemic therapies, enrollment in clinical trials, or tumor genomic profiling. The overall survival rate was not significantly different between the two populations in metastatic prostate, bladder, and kidney cancer, respectively. As available in a tertiary cancer hospital, access to care can mitigate the difference in the quality of healthcare and clinical outcomes in urban versus rural patients.

## 1. Introduction

Rural and urban disparities in cancer incidence and outcomes represent a significant issue in public health, particularly in the realm of oncology. Research suggests that despite substantial improvements in overall healthcare, a disparity in the burden of cancer exists between urban and rural populations [1,2,3]. Hashibe et al., for instance, showed a discernible difference in survival outcomes for cancer patients based on their geographical location. Specifically, they reported that rural cancer patients in Utah had a 5.2% lower five-year relative survival rate and a 10% increased risk of death compared to their urban counterparts [4]. Additionally, the disparities extend beyond survival rates to include other outcomes, such as stage at diagnosis and the availability of early detection and prevention programs.

This disparity in cancer outcomes can be partly attributed to disparities in access to high-quality healthcare. Rural communities frequently suffer from inadequate healthcare infrastructure, including a scarcity of specialized oncology care and comprehensive cancer centers [5]. This lack of access to appropriate care can delay the diagnosis and treatment of cancer, thus adversely impacting patient outcomes. Moreover, rural communities also face disparities in clinical research participation. Historically, rural patients have been underrepresented in clinical trials [6,7]. This lack of representation not only affects the applicability of trial results to this population but also limits access to novel and potentially more effective treatment strategies, which are typically available in the context of clinical trials. Recently, efforts to mitigate healthcare disparities have intensified, but these initiatives often lack the needed focus on the unique needs of rural populations. Although these efforts mark a positive shift, they have found it challenging to address the core problem: the geographical divide in access to top-tier cancer care. This underscores the crucial need for a more nuanced approach tailored to addressing the specific challenges faced by rural cancer patients.

As the landscape of cancer treatment continues to evolve, it is imperative that all patients, regardless of their location, have access to the highest quality of care [8]. Considering this, this study examines the implications of geographic access to high-quality healthcare services, such as those provided by Huntsman Cancer Institute in Utah. We focus specifically on advanced genitourinary (GU) malignancies, analyzing the impact of access on quality of care metrics and survival outcomes. GU malignancies—mainly comprising prostate, bladder, and kidney cancers—are associated with specific challenges in diagnosis, treatment, and patient outcomes. Focusing on this subset of cancers will enable a more nuanced understanding of rural–urban disparities and inform strategies for overcoming these barriers to equitable care.

## 2. Materials and Methods

### 2.1. Patients’ Selection

This retrospective study screened 2312 patients with advanced cancers who visited the GU oncology clinic at Huntsman Cancer Institute (HCI) over a three-year period (1 October 2017 to 30 September 2021). The presence of metastatic prostate cancer (mPCa), metastatic renal cell carcinoma (mRCC), or metastatic bladder cancer (mBCa) was required for further analysis. Patients with localized GU malignancies, adrenal gland tumors, testicular tumors, or other rare GU cancers were excluded from the study. There were 1025 patients that met the inclusion criteria and were included in the final analysis. The institutional review board approved the study.

### 2.2. Data Collection

Individual patient information was retrospectively extracted from electronic medical records. Included in the demographic data were primary residence zip codes, race/ethnicity, insurance status, age at diagnosis, and gender. The median household income data were obtained and calculated from “The Michigan Population Studies Center” (https://www.psc.isr.umich.edu, accessed on 1 September 2022) using a unique zip code. The study included the pathology type and de novo metastatic disease at diagnosis as the cancer-specific details. The metrics for measuring the quality of healthcare include the lines of systemic treatment received, genomic profile testing, accrual rates in clinical trials, and overall survival rate.

### 2.3. Statistical Examination

According to the patient’s residence, geographic variables were formulated. The residential addresses were initially geocoded and linked to the 2020 U.S. Census. A code for the rural–urban commuting area (RUCA) was assigned to each patient. RUCA is a classification scheme developed by the USDA that identifies each Census tract based on the percentage of urbanized residents from the U.S. Census and information on commuting flow [9]. Using the RUCA 4-tiered taxonomy (https://www.ers.usda.gov/data-products/rural-urban-commuting-area-codes, accessed on 1 September 2022), each patient’s residence was further classified as either urban (1–3) or rural (4–10) (https://www.ers.usda.gov/data-products/rural-urban-commuting-area-codes, accessed on 1 September 2022). The straight line distance between the patient’s home zip code and HCI was computed by assigning the locations latitude and longitude coordinates. The distance from HCI was divided into four categories: 10 miles, 10–50 miles, 50–100 miles, and >100 miles. Then, we compared the demographic variables (age, gender, race, distance from the cancer center) and cancer-related variables between urban and rural patients (tumor type and histological characteristics, lines of treatment received, clinical trial accrual rates, and frequency of genetic profile testing).

In this analysis, descriptive data were presented as means with standard deviation or medians with ranges. Pearson’s chi-square or Fisher’s exact tests were used to compare categorical variables. Using Kaplan–Meier methods, the overall survival rate was analyzed. To statistically compare the overall survival between groups, a logrank (Mantel–Cox) test was applied. All statistical analyses were conducted using SPSS 12.0. A two-sided *p*-value less than or equal to 0.05 was regarded as statistically significant.

## 3. Results

### 3.1. Demographics

The study screened a total of 2312 patients, and 1025 of them with metastatic GU cancer were included. Most of the patients (*n* = 860, or 83.9%) were from urban areas, while a smaller proportion (*n* = 165, or 16.1%) were from rural areas (Table 1). mPCa (*n* = 679, 66.2%) was the most prevalent form of cancer in our cohort, followed by mBCa (*n* = 184, 18.0%) and mRCC (*n* = 162, 15.5%). Most patients (*n* = 943, 92%) were White, followed by Hispanic/Latino (*n* = 43, 4.2%), Asian (*n* = 15, 1.5%), and African American (*n* = 9, 0.9%) (Appendix A). In all three subtypes of GU cancer, there was no statistically significance difference between the number of White patients in urban and rural areas [92.4% vs. 94.6%, *p* = 0.87 (mPCa); 95.6% vs. 100.0%, *p* = 0.88 (mBCa); 83.6% vs. 81.8%, *p* = 0.84 (mRCC)].

The combined financial data revealed a higher median household income of patients from urban areas ($64,604/year) than those from rural areas ($56,000/year). Compared to rural patients (Table 2), urban patients have a numerically lower proportion of Medicare coverage (44.9% vs. 50.9%) and a higher proportion of commercial coverage (40.4% vs. 35.1%) (Table 2). There is no difference in patients from urban and rural areas without insurance coverage (12.3% vs. 11.0%) and with Medicaid coverage (2.3% vs. 3.0%), respectively.

### 3.2. Accessibility to Genomic Testing and Clinical Trials

Overall, 30% of patients participated in clinical trials [mRCC (48.1%) *>* mPCa (28.3%) *>* mBCa (20%)]. Of the patients, 64.0% underwent genomic testing [mBCa (71.2%) *>* mPCa (65.4%) *>* mRCC (50%)] (Table 3). Regarding the recruitment of patients for clinical trials, there was no significant difference between urban and rural areas [27.3% vs. 33.3%, *p* = 0.22 (mPCa); 20.3% vs. 19.2%, *p* = 0.90 (mBCa); and 49.3% vs. 40.0%, *p* = 0.46 (mRCC)]. Similarly, there was no significant difference between urban and rural patients in terms of the frequency of genomic profiling tests [66.3% vs. 60.7%, *p* = 0.25 (mPCa); 71.5% vs. 69.2%, *p* = 0.72 (mBCa); and 50% vs. 50%, *p* = 1.0 (mRCC)] (Table 3).

### 3.3. Number of Treatment Lines Received

In this study, 42.0% received more than one line of therapy [mRCC (51.9%) *>* mPCa (42.1%) *>* mBCa (32.6%)] (Figure 1). In Figure 1, a darker color is correlated with a higher percentage of patients who received multiple lines of treatment. We investigated whether providing treatment beyond the first line differs in any way. There was no significant difference between urban and rural patients in the proportion of patients receiving more than one line of treatment [42.8% vs. 38.4%, *p* = 0.44 (mPCa); 32.9% vs. 30.8%, *p* = 0.99 (mBCa); and 53.6% vs. 40.0%, *p* = 0.27 (mRCC)] (Table 3).

### 3.4. Geographic Distribution of Patients

Most patients in our cohort (*n* = 588, 57.4%) from urban and rural areas lived within 10 to 50 miles of HCI (Appendix A). The percentage of the urban population among patients from 10 miles, 10–50 miles, 50–100 miles, and >100 miles away was 100%, 86.1%, 76.3%, and 35.4%, respectively (Appendix A). Patients’ data were analyzed to determine differences in access to care based on the distance between patients’ homes and the cancer center. No significant differences were observed in the clinical trial enrollment rates (*p* = 0.82), genomic profiling (*p* = 0.67), or the proportion of patients receiving more than one line of treatment (*p* = 0.29) based on distance (Appendix A).

### 3.5. Survival Outcomes

There was no statistically significance difference in the overall survival (OS) rate between the rural and urban populations. The median OS of patients from urban and rural areas was 72.7 and 95.8 months for mPCa (*p* = 0.14), 44.1 and 41.7 months for mRCC (*p* = 0.58), and 12.7 and 6.5 months for mBCa (*p* = 0.06), respectively (Figure 2).

## 4. Discussion

This study highlights an important factor in mitigating the disparity in healthcare outcomes between urban and rural cancer patients: access to a tertiary cancer hospital. Based on the analysis of 1025 patients with advanced GU cancers, we found that access to high-quality cancer care in a comprehensive cancer center can potentially bridge the divide between urban and rural patients in terms of healthcare quality and clinical outcomes. Notably, despite urban patients having a higher median household income and different insurance coverage patterns, the receipt of systemic therapies, enrollment in clinical trials, and tumor genomic profiling were comparable between the two populations. Likewise, overall survival did not differ significantly between the two groups, suggesting that access to high-quality care can yield comparable outcomes for urban and rural patients, thus alleviating geographic healthcare disparities.

Social determinants such as demographic location, ethnicity, racial background, socio-economic strata, education, and distance from tertiary health centers constitute a formidable barrier to healthcare. These social determinants have a significant impact on the cancer continuum and have been identified as one of the significant challenges requiring immediate attention [10,11]. This study found no difference in the frequency or histological subtype of GU cancers (prostate, bladder, or kidney) between urban and rural areas despite the lower socioeconomic status of rural patients (in terms of lower annual income and lower commercial insurance coverage). There was no statistically significant difference between urban and rural areas in terms of baseline mean PSA levels and Gleason scores among mPCa patients. Consequently, we discovered a uniform distribution of various GU cancer types, histological subtypes, and PSA levels/grade groups (for PCa) in both the urban and rural populations. Therefore, we hypothesize that the homogeneity of the cancer distribution pattern in our cohort nullified any cancer-specific factor that would otherwise be attributed to the disparity in clinical outcomes between patients from the two regions. We suggest that this equal accessibility to various diagnostic modalities (e.g., genomic profiling), treatment regimens (e.g., lines of treatment), and advanced treatment options (e.g., clinical trial recruitment) translated into similar survival outcomes for patients from rural areas as compared to urban areas. Our discovery is comparable to the most recent report by Unger et al. After providing patients from urban and rural areas with equal access to treatment in clinical trials, they found no significant difference in the survival outcomes between the two groups [8].

It is estimated that by 2030, the number of new cancer cases in the United States will increase by 45 percent compared to 2014. Therefore, it is crucial to ensure that all strata of cancer patients have equal and uninterrupted access to appropriate and advanced cancer care [12]. Recent estimates indicate that only 3 percent of oncologists practice in rural areas, where 20 percent of Americans reside. This oncologist–patient disparity is alarming and hinders rural populations’ access to equal cancer care [12]. Numerous factors, including race, level of education, gender, type of cancer, and place of birth, are immutable. The global effect of these unchangeable factors on cancer treatment and clinical outcomes is fixed and cannot be altered [13]. In this study, we hypothesize that by providing rural populations with equal access to a cancer center with tertiary care, the negative effects of non-modifiable factors traditionally associated with poor outcomes could be nullified or diminished. We discovered that patients treated at our cancer center had equal access to advanced tests such as genomic profiling and clinical trial enrollment. In addition, there was no significant difference in the mean number of systemic therapies received by patients from the two regions.

Multiple studies have demonstrated that rural patients are diagnosed with cancer at an older age and with more aggressive clinical characteristics. Overall, an advanced stage at diagnosis and restricted access to cancer treatment, including clinical trials, contribute to poorer prognoses [14,15]. Maganty et al. discovered that rural residents were less likely to undergo PCa treatment than urban residents across all risk categories, including low- [adjusted odds ratio (aOR):0.77], intermediate- [aOR:0.71], and high-risk disease [aOR:0.680] [16]. Both patient- and system-related factors contribute to late cancer diagnoses and subsequent treatment delays in rural populations. Among system-related factors, the absence of a comprehensive primary healthcare and cancer care system in or near rural areas can result in decreased screening, delayed referral to specialists, delayed diagnosis, and delayed treatment initiation [16,17]. By establishing new visiting consultation clinics for oncologists in rural communities, Ward et al. observed an increase in the rate of chemotherapy from 10% to 24% for newly diagnosed invasive cancers, thereby significantly enhancing local access [18].

Recent advances in cancer research and treatment have expanded patients’ treatment options beyond the standard of care. Multiple studies indicate that a higher level of education, private/public health insurance coverage, and urban residence increase the likelihood of participation in clinical trials [6,19,20]. Contrary to this conventional conclusion, our retrospective analysis reveals no significant differences between urban and rural populations in clinical trial enrollment rates. Even when stratified by distance from our cancer center, no statistically significant differences were observed. With increasing distance from our institute, a general upward trend in clinical trial enrollment rates was observed. This could likely be explained by the fact that many patients from farther away were referred for enrollment in clinical trials.

Genomic profiling of tumors provides a one-of-a-kind opportunity for a deeper understanding of tumor biology, disease prognosis/patient counseling, and the possibility of targeted therapy. In this regard as well, studies have uncovered a significant difference between the rural and urban populations regarding the use of newer diagnostic tools [21]. Salloum et al. found a significant awareness gap between rural and urban populations regarding direct-to-consumer genetic testing [21]. Contrary to this conventional conclusion, this study found no statistically significant difference between rural and urban patients in terms of genomic testing rates. Moreover, stratifying patients based on their proximity to our cancer center revealed no significant differences.

## 5. Limitation of the Study

We used the standardized RUCA four-tiered taxonomy to categorize our patients into a “rural” or “urban” region based on zip codes. Other studies, however, have concluded that the use of Census tract data is appropriate for analyzing differences in cancer outcomes [22]. Another potential limitation was the exclusion of localized GU cancers from this analysis. We limited this study to advanced GU malignancies as this is the most appropriate patient population reporting to oncologists. This study focused on the system-driven factors contributing to unequal access to cancer care. Hence, we excluded patient-related factors that are primarily non-modifiable, like level of education, marital status, performance status, etc., from this study.

## 6. Conclusions

Despite a higher median income and better insurance coverage in the urban population, there was no difference in quality of care metrics between urban and rural patients with advanced GU cancer. These findings indicate that access to quality care, such as that provided by a tertiary cancer hospital, can reduce the disparity in the quality of healthcare between urban and rural patients.

## Figures and Tables

**Figure 1 cancers-15-05171-f001:**
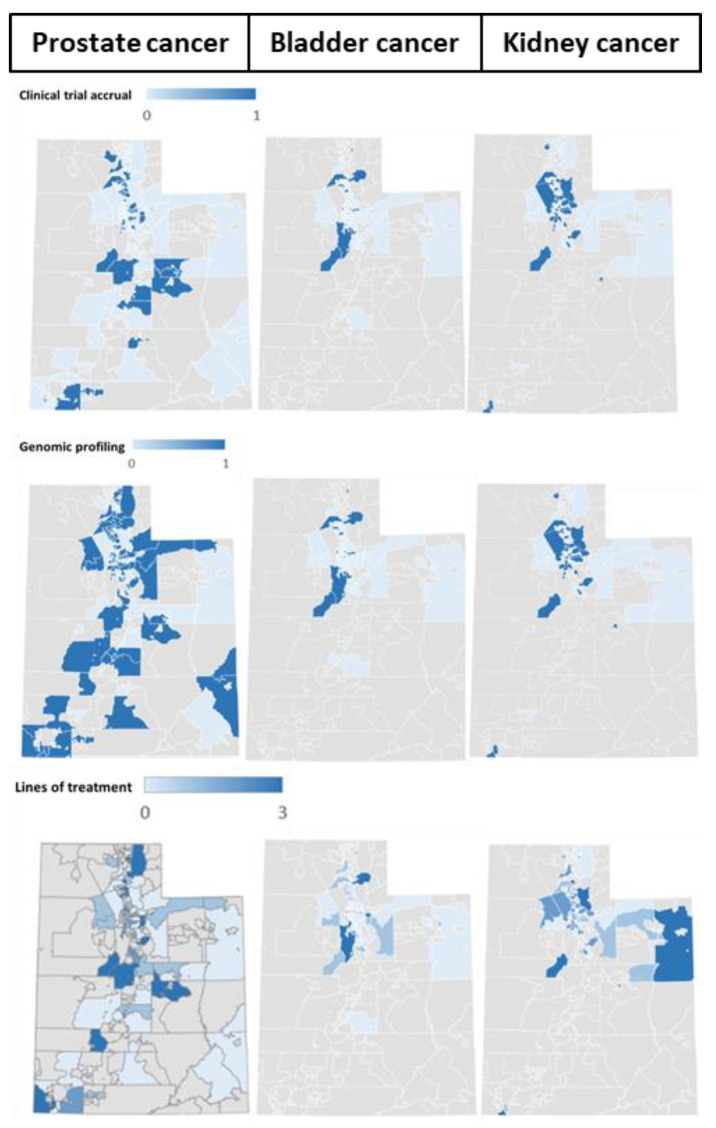
Distribution of patients by zip code for clinical trial enrollment (**top**), genomic profiling (**middle**), and lines of treatment (**bottom**) for prostate cancer (**left**), bladder cancer (**middle**), and kidney cancer (**right**).

**Figure 2 cancers-15-05171-f002:**
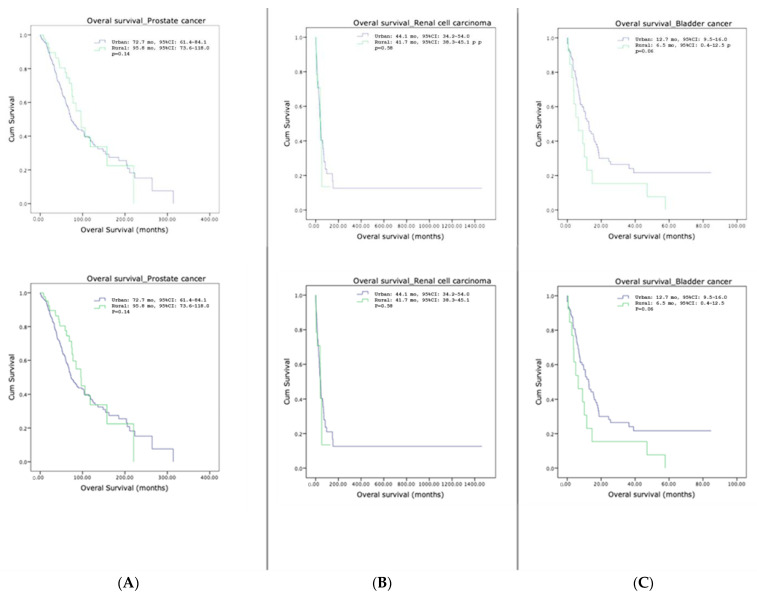
Overal survival of metastatic prostate cancer (**A**), metastatic renal cell carcinoma (**B**), and metastatic bladder cancer (**C**) in urban vs. rural areas.

**Table 1 cancers-15-05171-t001:** Baseline demographic and clinical characteristics of the patients.

		Urban	Rural	*p*-Value
Median age at diagnosis, years (range)	mPCa	65 (44–88)	65 (43–87)	0.84
	mBCa	66 (36–88)	69 (41–86)	0.22
	mRCC	60 (43–80)	59 (37–74)	0.56
Sex (male), *n* (%)	mBCa	114 (72.2)	17 (65.4)	0.64
	mRCC	99 (70.7)	15 (68.2)	0.81
Race (White), *n* (%)	mPCa	524 (92.4)	106 (94.6)	0.87
	mBCa	151 (95.6)	26 (100.0)	0.88
	mRCC	117 (83.6)	18 (81.8)	0.84
De novo metastatic disease at diagnosis, *n* (%)	mPCa	227 (43.0)	43 (40.6)	0.72
	mBCa	32 (20.3)	6 (23.0)	0.94
	mRCC	59 (42.0)	8 (36.4)	0.64
Pathology subtypes, *n* (%)	mPCa, adenocarcinoma subtype	550 (97.0)	111 (99.1)	0.34
	mBCa, urothelial subtype	85 (53.8)	17 (65.4)	0.37
	mRCC, clear cell subtype	105 (75.0)	14 (63.6)	0.25

**Table 2 cancers-15-05171-t002:** Insurance status of patients from urban and rural areas (*n* = 1025).

	Urban, (*n* = 860)	Rural, (*n* = 165)	*p*-Value
Insurance plan, *n* (%)			0.40
No insurance	106 (12.3)	18 (11.0)	
Medicare	386 (44.9)	84 (50.9)	
Medicaid	20 (2.3)	5 (3.0)	
Commercial	348 (40.4)	58 (35.1)	

**Table 3 cancers-15-05171-t003:** Health quality metrics comparison between patients from urban and rural areas.

		Urban, *n* (%)	Rural, *n* (%)	*p*-Value
Clinical trial accrual	mPCa	155 (27.3)	37 (33.0)	0.22
	mBCa	32 (20.3)	5 (19.2)	0.90
	mRCC	69 (49.3)	9 (40.9)	0.46
Genomic profiling	mPCa	376 (66.3)	68 (60.7)	0.25
	mBCa	113 (71.5)	18 (69.2)	0.72
	mRCC	70 (50.0)	11 (50.0)	1.00
Patients who received more than one line of systemic treatment	mPCa	243 (42.8)	43 (38.4)	0.44
	mBCa	52 (32.9)	8 (30.8)	0.99
	mRCC	75 (53.6)	9 (40.9)	0.27

## Data Availability

The data generated in this study are not publicly available as they could compromise patient privacy but are available upon reasonable request from the corresponding author.

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
