# Peer review of "Access to Care and Healthcare Quality Metrics for Patients with Advanced Genitourinary Cancers in Urban versus Rural Areas"

_cancers, 2023, doi:10.3390/cancers15215171_

Round 1
Reviewer 1 Report
Comments and Suggestions for Authors
Congratulations for the present study! The manuscrispt is well designed and written in a comprehensive manner.
There are few aspects that need to be improved.
1. Please exclude repetitions in your from the results section. You don't need to rewrite all the table data as long as they are already visible.
2. In Figure 1 please comment the different blue colors from the images.
3. Please try to avoid words like "Our study". It doesn't look too professional. You can use "This study" instead.
Author Response
- Please exclude repetitions in your from the results section. You don't need to rewrite all the table data as long as they are already visible.
Thanks for your feedback. In response to your first point, we have carefully reviewed the results section and removed any repetitive data, particularly from lines 138-142 and lines 148-153.
- In Figure 1 please comment the different blue colors from the images.
We appreciate your attention to detail. We've included comments in lines 175-176 to explain the different shades of blue in the images, ensuring clarity for the readers.
- Please try to avoid words like "Our study". It doesn't look too professional. You can use "This study" instead.
We acknowledge the preference for a more formal tone in the manuscript. We have made the necessary changes, replacing instances of "our study" with "this study" throughout the paper.
Reviewer 2 Report
Comments and Suggestions for Authors
In the manuscript entitled “Access to care and Health care Quality Metrics for Patients with Advanced Genitourinary Cancers in Urban Versus Rural Areas” authors have compared the retrospective data including household incomes, insurance status, accessibility to genomic testing and clinical trials and overall survival status of cancer patients from urban and rural areas. Interestingly, authors have found no significant difference in health quality metrics, healthcare accessibility, number of therapy treatment lines, and overall survival of cancer patients with advanced disease. There are several concerns about the data analysis, representation of the data and more importantly the explanation of the results as shown below:
Comments
# The major concern of this study is the enrollment of highly unequal number of cancer patients between urban (n=860) and rural (n=165) cohorts, that could lead to the potential bias in the differences in health quality metrics, survival, and other related data.
# In table 1, race-specific data related to Hispanic/Latinos, Asians and black patients are missing.
# What was the status of therapy treatment and disease outcomes according to insurance plan between Urban and Rural areas? A correlation analysis among these parameters would show how insurance plan could affect the health quality metrics and disease progression or outcome.
# In table 2, how was the P value calculated? Also please mention sample size (n) in both the cohorts (urban and rural).
# Was there any correlation between the distance of cancer center from patients’ residence and overall survival of patients between urban and rural areas?
# Some of the sentences in the text are incomplete and there are some grammatical and typo errors. I strongly encourage authors to critically read the manuscript.
Comments on the Quality of English LanguageSome of the sentences in the text are incomplete and there are some grammatical and typo errors. I strongly encourage authors to critically read the manuscript.
Author Response
# The major concern of this study is the enrollment of highly unequal number of cancer patients between urban (n=860) and rural (n=165) cohorts, that could lead to the potential bias in the differences in health quality metrics, survival, and other related data.
Thank you for commenting on the unequal number of patients between the urban and rural cohorts. We think the main reason is the demographic realities of the state of Utah. The state has seen significant urbanization in recent years, with most of its population residing in urban centers, particularly along the Wasatch Front, which includes cities like Salt Lake City, Provo, and Ogden. There are a number of other reasons (please see below).
Firstly, using RUCA (Rural-Urban Commuting Area) codes is a standard method to determine rurality, which has been widely adopted in various studies. Many other studies have used RUCA codes to determine healthcare access disparities based on geographical classifications ([Reference: Castle ME, Tak CR. Self-reported vs RUCA rural-urban classification among North Carolina pharmacists. Pharm Pract, 2021). It is a recognized observation in the literature that urban populations tend to be larger than rural populations. By definition, urban areas are characterized by higher population densities and vast human features in comparison to areas surrounding them, resulting in naturally higher populations in urban areas.
Therefore, while we acknowledge the disparity in the sample size between our urban and rural cohorts, this distribution reflects the real-world demographic distribution and is consistent with prior research.
# In table 1, race-specific data related to Hispanic/Latinos, Asians and black patients are missing.
Thank you for pointing out the omission of race-specific data for Hispanic/Latinos, Asians, and black patients in Table 1. To address this, we have included a detailed breakdown of the data for these racial groups in the supplementary section. Please refer to Table S1 in the supplementary materials for this comprehensive data. We appreciate your attention to detail, ensuring that our study remains as thorough and inclusive as possible.
# What was the status of therapy treatment and disease outcomes according to insurance plan between Urban and Rural areas? A correlation analysis among these parameters would show how insurance plan could affect the health quality metrics and disease progression or outcome.
Regarding the relationship between insurance plans and healthcare outcomes, we had another study looking into this. The preliminary results are mixed, with no clear relationship between insurance plan and disease outcomes in GU cancer patients.
# In table 2, how was the P value calculated? Also please mention sample size (n) in both the cohorts (urban and rural).
It was calculated using Chi-square test. The method was explained in the method section. The sample size was added to each cohort.
# Was there any correlation between the distance of cancer center from patients’ residence and overall survival of patients between urban and rural areas?
We calculated and presented the distance of the cancer center from the patient’s residence in Table S2 and Figure S1. There was no correlation between the distance and overall survival.
# Some of the sentences in the text are incomplete and there are some grammatical and typo errors. I strongly encourage authors to critically read the manuscript.
We have carefully read through the manuscript and corrected those typographical errors.
Round 2
Reviewer 2 Report
Comments and Suggestions for Authors
Authors have addressed all the comments satisfactorily. No further comments are needed.